# Neutrophil Heterogeneity in Wound Healing

**DOI:** 10.3390/biomedicines13030694

**Published:** 2025-03-12

**Authors:** Filippo Renò, Corinna Anais Pagano, Monica Bignotto, Maurizio Sabbatini

**Affiliations:** 1Health Sciences Department (DiSS), San Paolo Hospital, Università di Milano, Via A. di Rudini 8, 20142 Milano, Italy; filippo.reno@unimi.it (F.R.); monica.bignotto@unimi.it (M.B.); 2Department of Science and Innovation Technology (DISIT), Università del Piemonte Orientale, Via T. Michel 11, 15121 Alessandria, Italy; corinnaanais.pagano@uniupo.it

**Keywords:** neutrophils, aging, skin, innate immunity, inflammation, plasticity

## Abstract

Neutrophils are the most abundant type of immune cells and also the most underestimated cell defenders in the human body. In fact, their lifespan has also been extensively revised in recent years, going from a half-life of 8–10 h to a longer lifespan of up to 5.4 days in humans; it has been discovered that their mechanisms of defense are multiple and finely modulated, and it has been suggested that the heterogeneity of neutrophils occurs as well as in other immune cells. Neutrophils also play a critical role in the wound healing process, and their involvement is not limited to the initial stages of defense against pathogens, but extends to the inflammatory phase of tissue reconstruction. Neutrophil heterogeneity has recently been reported at the presence of distinct subtypes expressing different functional states, which contribute uniquely to the different phases of innate immunity and wound healing. This heterogeneity can be induced by the local microenvironment, by the presence of specific cytokines and by the type of injury. The different functional states of neutrophils enable a finely tuned response to injury and stress, which is essential for effective healing. Understanding the functional heterogeneity of neutrophils in wound healing can unveil potential pathological profiles and therapeutic targets. Moreover, the understanding of neutrophil heterogeneity dynamics could help in designing strategies to manage excessive inflammation or impaired healing processes. This review highlights the complexity of neutrophil heterogeneity and its critical roles throughout the phases of wound healing.

## 1. Introduction

Neutrophils are the predominant immune cells in human blood; they patrol and protect the host from pathogens and other harmful agents. Their involvement is observed in inflammatory sites, tumor sites or any area of the body where various types of injuries occur. Although some neutrophils can occasionally be tissue-resident due to their patrolling activity, during inflammation they are extensively recruited from circulation, crowding the inflamed site and initiating their defensive role [1,2]. The essential importance of neutrophils in the clearance of infection is highlighted by experimental and clinical data showing that the localization of neutrophils to the site of inflammation is crucial.

Neutrophils can eliminate pathogens through multiple mechanisms, such as phagocytosis, degranulation, reactive oxygen species (ROS) production and the formation of neutrophil extracellular traps (NETs) [3]. Indeed, a marked decrease in neutrophil concentration in the blood leads to severe immunodeficiency in humans [4]. Neutrophils have long been described as a homogeneous population of immune cells, but their uniformity has been questioned. A new perspective has emerged, revealing different functions and possibly different roles in various conditions expressed by neutrophils [5,6].

Indeed, the heterogeneity of neutrophils is a previously discussed finding, although the real importance of this finding has not been adequately assessed or understood [7].

The lifespan of circulating neutrophils has also been extensively revised, and the previous identification of these cells as short-lived, with a reported half-life of 8–10 h, has been updated, with estimates suggesting a much longer lifespan of up to 5.4 days in humans [2].

Neutrophils are required for their defensive action in wounds, where they participate in both the inflammatory and regenerative phases of wound healing together with macrophages and lymphocytes [2]. In fact, during wound healing, neutrophils are the most numerous group of cells to be recruited from the circulation in the wound site, and the remnants are then cleared by macrophages [2,5].

Recently, the relationships between cancer cells and neutrophils have gained wide interest, since it has been clearly demonstrated that different neutrophil populations can exert positive and negative effects in terms of tumor progression and drug resistance [8,9,10]. Therefore, neutrophil heterogeneity could also play a role in other physiological or pathological phenomena.

The scope of the present review is to illustrate the current knowledge about neutrophil heterogeneity and its possible role in both normal and altered wound healing.

## 2. Neutrophils in Wound Healing: State of the Art

The wound healing process is classically organized into four overlapping stages named hemostasis, inflammation, proliferation and remodeling [11,12].

Hemostasis begins immediately after injury with the narrowing of the damaged vessels, followed by platelet aggregation. The platelets release a plethora of growth factors, stimulating the migration and proliferation of local tissue cells [13]. During inflammation, increased capillary permeability allows for the rapid recruitment of neutrophils to the wound site, driven by various chemotactic factors [2,14]. Once at the site, the neutrophils utilize ROS and reactive nitrogen species to neutralize pathogens, along with the formation of neutrophil extracellular traps (NETs) and the release of antimicrobial substances such as elastase and myeloperoxidase [15] (Figure 1A). Neutrophils are also involved in tissue remodeling, as they help clear necrotic tissue by producing and releasing matrix metalloproteinases (MMPs), particularly MMP-8 and MMP-9, as well as other proteolytic enzymes like elastase and cathepsin G. This activity contributes to the degradation of damaged extracellular matrix components [16,17]. This proteolytic activity is essential to facilitate the subsequent repair processes, allowing for the migration of other cell types involved in tissue regeneration. Overall, neutrophils serve both as defenders against infection and as facilitators of tissue repair in the early stages of inflammation (Figure 1A). Their timely recruitment and functional activities are vital to restoring homeostasis following injury [1]. Other defensive cells are also recruited to the wound site with different roles. For example, macrophages are recruited early to phagocytose microorganisms, and later to scavenge debris and to replenish the extracellular matrix [18]. In particular, during wound healing, a short-lived type of macrophage called M1 is immediately recruited to eradicate invading microorganisms and promote type I immune responses. In the later phase, a long-lasting type of macrophage, called M2, emerges as a result of the polarization of some macrophages by the local environmental stimuli. M2 macrophages promote debris scavenging, resolution of inflammation, angiogenesis and tissue remodeling [18].

Lymphocytes are attracted to the wound site largely by cytokines secreted by the macrophages and neutrophils. These lymphocytes contribute to the adaptive immune response, enhancing the overall defense against any remaining pathogens [18] (Figure 1B).

Keratinocytes, the predominant cell type in the epidermis, also participate in host defense by producing antimicrobial peptides (AMPs) such as human β-defensin [19] and cathelicidin (hCAP18/LL-37). These peptides possess direct antimicrobial properties and also act as chemotactic agents to recruit inflammatory cells to the wound site. Furthermore, keratinocytes express toll-like receptors (TLRs), which recognize pathogen-associated molecular patterns and mediate the recruitment of immune effector cells [20]. Macrophages were considered the key cells of the reconstructive phase of the matrix tissue, but experimental evidence in a murine model has reported that macrophage depletion during the late stage of healing does not seem to affect scar tissue formation [21] (Figure 1C). This finding has drawn attention to the neutrophils’ role during the remodeling phase. In fact, MMP-9 is involved in the breakdown of extracellular matrix components and promotes the endothelial cell migration and proliferation assisting in the process of forming new blood vessels [17], and neutrophils are the only cells capable of releasing MMP-9 from its endogenous tissue inhibitor of metalloproteinase-1 (TIMP-1), delivering the highly active MMP-9 to angiogenic sites [17]. Indeed, neutropenic patients have difficulty with wound healing, and although this observation awaits more direct confirmation, it is clear that neutrophils are key cells in efficient wound healing [22].

During the development of wound healing, a large number of neutrophils do not die during the defensive process, but undergo apoptosis, facilitated by activated macrophages [2]. Moreover, the occurrence of the reverse migration of exhausted neutrophils towards the bone marrow has also been observed, both in patients [23] and in experimental models [1,24] (Figure 1C).

The proliferation phase is characterized by fibroblasts and endothelial cells that undergo a high rate of proliferation due to the presence of macrophage-secreted growth factors such as Platelet-Derived Growth Factor (PDGF) and Insulin-Like Growth Factor-1 (IGF-1) [25]. Also, neutrophils seem to actively participate in the proliferation phase, secreting vascular endothelial growth factor (VEGF) to stimulate the initial angiogenesis [26].

Moreover, neutrophils have also been observed to stimulate epithelial cells to secrete VEGF [1,26]. During this phase, keratinocytes proliferate and migrate to restore the epidermal barrier, in turn secreting growth factors such as keratinocyte-derived anti-fibrogenic factor (KDAF), which initiates the scar tissue remodeling phase [27].

The last phase of wound healing consists of granular tissue transformation into scar tissue. During this phase, some of the fibroblasts differentiate into myofibroblasts. These cells interact with the extracellular matrix (ECM) via the main integrin receptors α_1_β_1_ and α_2_β_1_, which allow for tissue contraction [28]. Then, further matrix remodeling occurs; classical evidence has pointed to M2 macrophages as being responsible for the remodeling phase, but recently, the more direct involvement of neutrophils has been highlighted [1,29].

All of these recent findings clearly indicate that neutrophils actively participate in all phases of wound healing.

## 3. Neutrophil Heterogeneity

Neutrophils have traditionally been considered a relatively homogeneous cell population with a primary role in the innate immune response [30], particularly in fighting bacterial infections, but accumulating evidence supports the hypothesis that distinct neutrophil subsets, each with different roles in infection, inflammation and cancer immunology, exist [5,14,31,32,33,34,35].

Neutrophil heterogeneity refers to the variability or diversity in their morphological characteristics and cellular behaviors. It is still an open issue whether these subsets represent truly distinct lineages or instead develop from a single plastic neutrophil precursor [1], although the linear development of some types of neutrophils has been observed [36]. Actually, the mainstream hypothesis considers the heterogenous neutrophil population to be composed of a single cell type that encompasses many different functional phenotypes [34,37]; therefore, the heterogenous neutrophil functional response is most likely to be context-dependent, where the inflammatory response, tissue environment and interaction with other cell types can induce specific and differentiated neutrophil response profiles. In fact, during the last decade, the existence of neutrophil subsets in different models has been demonstrated. Some of these subsets appear to be disease- or tissue-specific, while the factors that govern the generation of heterogeneity are largely uninvestigated [38]. Few of these subsets have been identified based on their injury or repair functions. Furthermore, authors have highlighted that human circulating neutrophils are transcriptionally dynamic cells, developing from a pre-mature state towards one of two distinct transcriptional phenotypes that cannot be adequately defined by common surface markers [39].

Following this, we analyze several distinct key features that allow us to identify the neutrophil subsets (the findings are summarized in Table 1).

### 3.1. Neutrophil Activation

Neutrophils can adopt different functional states depending on their activation and the signals they receive from the microenvironment [40]. Neutrophils become typically activated by damage-associated molecular patterns (DAMPs) in the case of trauma or by pathogen-associated molecular patterns (PAMPs) upon encountering pathogens, showing changes in surface markers, increased production of ROS and the release of inflammatory cytokines (Figure 1A) [1,41].

The response of the neutrophils upon stimulation by the cytokines is regulated by vesicle release. In fact, a large part of the secretory vesicle protein (secondary and tertiary granules) represents a reservoir of plasma membrane proteins that can be rapidly incorporated into the cell surface upon neutrophil activation, inducing neutrophil adhesion, extravasation and functionality [34].

More recently, the existence of primed neutrophils has been recognized, represented by neutrophils exposed to sub-lethal signals that enhance their responsiveness to subsequent challenges [42,43] (Figure 2). Priming and partial activation are concomitant with the release of tertiary granules, preparing the neutrophils to be fully activated in response to specific agonists [34]. Integrins play a primary role in this process, as integrin engagement serves to enhance neutrophil functions [44].

Neutrophils acting at an inflammatory site cannot be considered dead-end cells. Even if the majority of them will die shortly following the defensive/immunoactive events and will be drained from the site by macrophages [45], several neutrophils will re-enter the circulation [46]. We indicate this subtype of neutrophils as “veteran neutrophils”, i.e., neutrophils that have been actively involved in defense mechanisms and remain alive, and that have not been induced to apoptotic death.

After reverse migration in the vascular blood, these “veteran neutrophils” will be drained by the spleen to be destroyed or they will re-enter the bone marrow, where they are supposed to influence the training of pro-myeloid cells via epigenetic and metabolic mechanisms, maturing neutrophils that will exhibit a hyperreactive phenotype [47] (Figure 2). This escape phenomenon has also been associated with a reversion of the “veteran neutrophil” phenotype from a terminal immunoactive state to a pro-inflammatory state, which has also been associated with dissemination events of systemic inflammation [23,46].

Finally, along with the activated, primed and “veterans neutrophils”, marginated and tissue-resident neutrophils are crucial and often underscore elements in the first responses to infections and tissue injuries [48,49].

### 3.2. Neutrophil Surface Markers

Neutrophils can show variability in the expression of surface markers and molecules that influence their functions. These differences can be triggered by the tissue microenvironment, infection status and presence of other immune cells [50].

A distinct population of immature bone marrow neutrophils has been observed in cancer patients, the so-called “low-density” neutrophils (LDNs) (Figure 2) [50]. LDNs inhibit T and NK cell proliferation [51], and they can be separated by density gradient along with monocytes. They have been termed as granulocytic/neutrophilic myeloid-derived suppressor cells (G/N-MDSCs). More specifically, a study by Pillay and coll. [52] has identified, within the MDSC population, a group of neutrophils expressing the membrane factor CD62L. This population suppresses T lymphocytes by releasing ROS, operating in response to systemic inflammation in patients. These T cell suppressor neutrophils were documented to expand in tumor-bearing hosts compared to healthy subjects [38]. In cancer patients, GMDSCs are CD11b^+^-, CD14^−^-, CD66b^+^- and CD15hi-expressing cells that are enriched in LDN fractions present within peripheral blood mononuclear cells (PBMCs) [38,53].

In a previous work, Pillay and coll. [54] have identified another group of neutrophils secreting the immunosuppressive cytokine IL-10 following bacterial and fungal infections. In the inflammatory response, the activity of the macrophages is dampened by the presence of a large amount of IL-10 [55]; therefore, it is possible that this new subset acts as control cells against excessive macrophage activation.

In the response to microbial infection, neutrophils are able to activate a particular response by extruding NETs composed of chromatin and antimicrobial proteins to trap and kill pathogens, but this response is not activated indifferently by all neutrophils in every condition. A specificity of response has been identified, depending on the external agent and probably on the site where the immunity defense take place [56]. In particular, the expression of the surface protein olfactomedin-4 (OLFM4) has been found associated with the neutrophils selectively releasing NETs, providing the opportunity to identify this particular subset of neutrophils [57]. NET release (NETosis) occurs following two different pathways.

In the so-called “suicidal NETosis”, the defensive activation of neutrophils leads the enzymes elastase and myeloperoxidase, present in the granules, to lyse parts of the nuclear and plasma membranes, unfolding chromatin and releasing it into the cytosol and, following the rupture of the plasma membrane, into the extracellular space, causing the death of the cell [58]. The “living NETosis” is instead activated by the interaction of the toll-like receptors on the platelet membrane and the CD11a ligands present on the neutrophils, causing nuclear budding and the release of NETs through the vesicles. This event leads to the cells remaining alive, able to continue phagocytic functions [58]. Together, these two alternative pathways may identify further neutrophil subsets.

Another molecule whose expression defines a particular group of circulating neutrophils is the surface glycoprotein CD177. Its expression has been found increased in patients with several bacterial infections, associated with the neutrophil surface proteinase 3 (PR3) expression. The function of the CD177–PR3 complex is not fully understood, but a recent study shows its potential role as a modulator of neutrophil recruitment from the circulation. In fact, CD177 expresses binding properties to tissutal molecules, and its association with PR3 may support increased neutrophil tissue infiltration [59].

An interesting and somewhat puzzling observation is the identification of a group of neutrophils (about 5–8% of the population) expressing T cell receptors (TCRs) αβ at their cell surfaces [60]. Binding to the TCR complexes present on the neutrophils results in protection from apoptosis, and it is involved in the stimulation of IL-8 secretion [59].

A further group of “proangiogenic neutrophils” has been identified by the expression of VEGF-A receptors, (VGFR1) receptors and VLA-4 integrin (CD49d/CD29, α4/β1) [61,62]. This neutrophil subset is able to penetrate hypoxia sites facilitating vessel growth, highlighting the intervention of neutrophils in tissue remodeling after tissue damage.

### 3.3. Neutrophil Cytokines

Neutrophil heterogeneity was first identified in 1986 when, following a gradient separation procedure, low-density neutrophils (LDNs) and high-density neutrophils (HDNs) were separated. These two populations were easily identifiable in the blood from acute and/or chronic inflammatory disease patients. The LDN line can be detected within the mononuclear cell fraction of the PBMCs, while the HDNs can be detected within the granulocyte fraction on top of the red blood cell layers.

The LDN exhibits elevated levels of pro-inflammatory cytokines, including type I interferons (IFNs), tumor necrosis factor (TNF) and interferon-gamma (IFN-γ), but LDNs demonstrate a compromised ability to phagocytize pathogens compared to other non-LDN neutrophils [53,63]. In the LDN fractions of healthy individuals, a high presence of mature cells with a primed phenotype has been observed [43].

Systemic lupus erythematosus (SLE) is a complex autoimmune disease characterized by the production of autoantibodies against nuclear components, leading to inflammation and damage in various tissues. It has been indicated that approximately 17% of the PBMCs in the bloodstreams of SLE patients consist of LDNs [64,65]. Emerging evidence indicates that LDNs play a significant role in the lupus pathogenesis through the process known as NETosis, as LDNs can spontaneously undergo NETosis [64,65]. The excessive formation of NETs has broader implications for the autoimmune processes in SLE. Increased NET release leads to the generation of NET-derived immune complexes that can activate plasmacytoid dendritic cells, resulting in the release of IFN-alpha (IFN-α). This cycle creates a positive feedback loop in which the IFN- α release stimulates further NET formation in the normal neutrophils from lupus patients, exacerbating the inflammation and tissue damage associated with SLE [64,65,66]. The interplay between the LDNs, NETosis and immune activation underscores the complexity of the immunological landscape in SLE and highlights potential therapeutic targets for intervention in this challenging disease.

The HDN fraction is instead characterized by the presence of active neutrophils with high granule contents, and enhanced pro-inflammatory activity has been observed [34,41]. Recently, this HDN cell population being addressed as “standard neutrophils” has been questioned, and actually, a different categorization is preferable, considering and calling high-density neutrophils as normal neutrophils [67] (Figure 2). Neutrophils have been found to be able to promote monocyte and macrophage recruitment by secreting LL37 and azurocidin [2].

In mice, two different subsets of neutrophils with specific functions and markers have been identified in cancer and inflammation studies. The two neutrophil subsets have been named N1 and N2, similarly to what is used for the macrophage subsets M1 and M2. In fact, N1 identifies a neutrophil subset showing pro-inflammatory and anti-tumor properties, while N2 neutrophils are associated with tissue repair and tumor promotion [1,68]. Differing interleukin expression has been observed in N1 and N2 neutrophils recruited in tissue hypoxia following myocardial infarction [68]. In the hypoxic site, the N1 population is firstly recruited, expressing IL-1β, IL-6 and TNF-α, and is then substituted by the N2 population, expressing arginase-1 (Arg1), Il-10 and Ym1 (rodent-specific chitinase-like protein, CLP) [69]. This shift in the neutrophil subsets is induced by tissue pro-inflammatory factors for the N1 subtype and by the appearance of tissue anti-inflammatory factors for the N2 type [69], highlighting a cellular polarization that could identify two phases of neutrophil intervention in the defensive and reconstructive phases of tissue damage. However, in humans, this specific simplified subsetting has not been observed.

### 3.4. Senescent Neutrophils

Neutrophils that remain in the vasculature undergo senescence, becoming dysfunctional and no longer effective in innate immunity, and they must be cleared from the blood [70] (Figure 2).

Senescent neutrophils are cleared by relocation to the bone marrow, thanks to increased expression of C-X-C chemokine receptor type 4 (CXCR4), a receptor for stromal cell-derived factor-1 (SDF-1), largely expressed in bone marrow [70]. This process is also supported by a reduction in CXCR2-mediated responses, which makes the senescent neutrophils less responsive to chemokines and less able to migrate to sites of inflammation [71]. The other neutrophils that do not undergo senescence (activated neutrophils) become apoptotic or are cleared from the circulation by the spleen (Figure 2).

In conclusion, neutrophils are not entirely fixed in their functions and can exhibit a form of plasticity. They can adapt and change depending on their environment, becoming a highly dynamic part of the immune system. This plasticity is part of what makes neutrophil heterogeneity a topic of considerable interest in immunology.

**Table 1 biomedicines-13-00694-t001:** Neutrophil heterogeneity on the basis of several defense parameters.

Classification	Type	Neutrophils	References
Activation		PAMP/DAMP-activated	[1,41]
		Primed	[42,43]
		Reverse-circulating	[46]
		Tissue-resident	[48,49]
Surface Markers	Low-Density Neutrophils (LDNs)	G/N-MDSCs—adaptive immune suppressor cells	[51]
		(G/N-MDSCs)—CD62L^+^–T cell suppressing	[38,52]
		(G/N-MDSC)—CD11b^+^, CD14^−^, CD66b^+^, CD15hi-suppressor cells in cancer	[38,53]
		IL-10 secreting–macrophage dampening	[54]
	High-Density Neutrophils (HDNs)	OLFM4—suicidal NETosis	[57,58]
		OLFM4—living NETosis	[58]
		CD177-PR3—tissue infiltration membrane factors	[59]
		TCRs-αβ—apoptotic protective and IL-8 secretion	[60]
		VGFR1/VLA-4—proangiogenic neutrophils	[61,62]
Cytokines	LDNs	IFN; TNF; IFN-γ—low phagocytic activity	[53,63]
	HDNs	LL37; Azurocidina—high phagocytic activity	[2,34,41,67]
	N1 (mouse)	Pro-inflammatory/anti-tumoral Il-1β; Il-6; TNF-α	[1,68,69]
	N2 (mouse)	tissue repair/tumor promotion Arg1; Il-10; Ym1	[1,68,69]
Senescence		CXCR4^high^/CXCR2^low^—bone marrow relocating	[70,71]

### 3.5. Metabolic Changes Supporting Neutrophil Heterogeneity

Long believed to be uniquely dependent on glycolysis, neutrophils have recently exhibited significant metabolic plasticity, adapting their energy production and biosynthesis in response to microenvironmental cues [72]. This metabolic plasticity allows neutrophils to exploit extracellular proteins and other substrates to sustain central carbon metabolism beyond glucose-derived pathways [72].

In general, neutrophils can switch between glycolysis and oxidative phosphorylation, depending on their activation state, to facilitate phagocytosis and ROS production [73]. Under high-activity conditions, glutamine may represent an alternative fuel used by neutrophils to increase oxidative metabolism (OxPHOS). Fatty acid oxidation (FAO) also plays a role in supporting neutrophil survival and function in chronic inflammatory conditions, especially under low glucose availability [74]. On the contrary, hyperglycemia has been shown to increase the formation of NETs [75] in an unstable form and poor in antimicrobial peptides in diabetes mellitus patients [76].

A particular subtype of LDN, known as tumor elicited neutrophils (TENs), has been shown to utilize the catabolism of glutamate and proline to activate mitochondrial-dependent ATP production and perform NETosis under glucose-deprived conditions, promoting extensive liver metastasis [77].

Finally, in tissue-transmigrating neutrophils, increased mitochondrial FAO- OxPHOS is essential to increase ATP production [78].

In conclusion, this diverse array of metabolic changes further supports the heterogeneity of neutrophils and their ability to express different cell subtypes in response to different physiological and pathological conditions.

### 3.6. Transcriptional Factors Involved in Neutrophil Heterogeneity

The role of transcriptional factors (TFs) during granulocytopoiesis has been extensively investigated [79], while their involvement in neutrophil subsetting has not been clarified. For example, NF-kB, a ubiquitous transcription factor involved in the inflammatory and immune responses and able to regulate the expression of genes that control cell survival and proliferation, may be a key factor in neutrophil subsetting. In fact, neutrophils exposed to pro-inflammatory stimuli such as TNF-α undergo the phosphorylation of IKK kinases simultaneously with the degradation of inhibitory IKK- α, both at the cytoplasmic and nuclear levels. These events induce the activation of NF-kB and start the inflammatory response [80]. Increasing evidence indicates that alternative NF-kB activation could otherwise inhibit inflammation [79]. From this perspective, a balance between the two NF-kB activation pathways in neutrophils could modulate the intensity and extent of the inflammatory phase during wound healing.

MAPK subtype cascades are another key set of signaling pathways mediating neutrophil activation. MAPK subtype cascades are activated upon the stimulation of G- protein coupled receptors (GCPRs) [79]. Human neutrophils express all three MAPK subtypes, including the ERK, c-Jun terminal kinase (JNK) and p38 MAPK cascades.

Different types of stimulatory agents induce the selective activation of the distinct MAPK subtype cascades, such as G-CSF, which exclusively activates the MEK-ERK cascade [79].

The activation of the p38, MEK/ERK and JNK cascades regulates a plethora of transcriptional factors (e.g., c-Jun, JunB, ATF2, p53, c-Myc) that regulate the transcription of several genes involved in morphological modification, inflammatory responses, survival and apoptosis upon neutrophil activation [79].

CAMP responsive element binding protein 1 (CREB1) is a transcription factor activated by MAPKs that plays a crucial role in neutrophil inflammatory responses. Human resting neutrophils strongly express CREB1, and upon their activation, the CREB-1/ATF-1 (activating transcription factor-1) complex enhances pro-inflammatory cytokine production [81].

CREB proteins are also rapidly phosphorylated by p38 MAPK kinases, which mediate the release of the chemoattractants CXCL8 and CCL4 and the expression of matrix metallopeptidase 9 (MMP9), strongly supporting the involvement of CREB1 in regulating neutrophil pro-inflammatory functions. CREB1 is strongly expressed by resting neutrophils, indicating its role in determining this subtype of neutrophils [79].

Clock-related TFs are proposed to be responsible for physiological neutrophil aging [82]. Neutrophil aging has been shown to follow a strict diurnal pattern characterized by a specific change from the membrane expression of high CXCR2, low CXCR4 and high CD62L to that of low CXCR2, high CXCR4 and low CD62L. In particular, brain and muscle aryl hydrocarbon receptor nuclear translocator-like (Arnt) 1 (Bmal1) regulates the aging process of neutrophils by modulating the expression of CXCL2, a CXCR2 ligand that functions to promote the process in an autocrine manner [79]. Transcriptionally, Bmal1 interacts with Clock to form a transcriptional activator complex that regulates the cell-intrinsic rhythms of the neutrophil diurnal aging process. However, more recently, oscillations in the expression of adhesion molecules in humans have been proposed to alternatively regulate the diurnal aging of neutrophils [83]. Since the adhesion molecules can regulate the redistribution and function of the circulating neutrophils in the body, they can also determine the rate of the aging process.

## 4. Neutrophil Heterogeneity in Wound Healing

### 4.1. Physiological Wound Healing

Neutrophils play a crucial role in the immune response during different stages of wound healing. They are typically the first responders to sites of injury or infection, where they contribute to both inflammation and tissue repair. The different stages of wound healing have a profound impact on the neutrophil activations and responses as they adapt to the evolving and plastic healing program (Figure 1).

The preliminary arrival of neutrophils at the site of inflammation is essential to bring an adequate number of cells for the subsequent defensive actions. The neutrophils adopt several mechanisms to bypass the endothelial lining of the capillary [84].

Integrins assume a fundamental role, both in diapedesis and also as co-stimulatory molecules, contributing to an initial phase of cell priming that enhances neutrophil effector functions [44].

During the inflammatory phase of wound healing, neutrophils are the most promptly recruited cells. The first classical action of neutrophils consists of highly phagocytic functions and the production of large amounts of ROS, playing a key role in clearing pathogens in the early inflammatory phase. Some neutrophils undergo a process called NETosis, in which they release chromatin fibers (NETs) to trap and kill pathogens [41,84]. However, high levels of NETs have been reported to cause toxicity to host cells [85] by the cytotoxic effects of histonic proteins [86]. Nevertheless, a low level of inflammation and NET production seems to promote keratinocyte proliferation via NF-kB [87].

Phagocytic action, ROS production and NET production identify several neutrophil subtypes whose specific actions are modulated by environmental stimuli (Figure 1B).

Although neutrophils are traditionally considered as immediate active defensive cells, now, some evidence indicates that neutrophils interact with various local immune cells.

This aspect reveals a further novel subtype of neutrophils, in which their relationships with other immune cells and the production of cytokines characterize their specific functions. Neutrophils directly induce the maturation of antigen-presenting cells (APCs) [88]. In particular, neutrophils can directly interact with dendritic cells (DCs) and promote their maturation by releasing pro-inflammatory cytokines, such as tumor necrosis factor-α (TNF-α) and IL-12, which enhance antigen presentation and the subsequent activation of T cells [89,90]. These findings pose neutrophils as a bridge between the innate and adaptive immune responses [84]. In fact, neutrophils also act as regulatory cells, as they can acquire or express regulatory properties that help modulate the immune response and support wound healing. Furthermore, neutrophils and NETs promote cytokine production by macrophages [91,92], and they are able to prime T lymphocytes [93] (Figure 1).

Neutrophils and neutrophil-derived cytokines promote T-cell cytokine production and differentiation, inducing bidirectional effects and functional interferences [94], and also trigger B-cell expansion and antibody production [84]. In this context, neutrophils become the essential agents in promoting the release of cytokines to support the inflammatory process by activating and maintaining both processes on the sides of the innate immune response and the adaptive immune response. More precisely, the neutrophils shift and support the balanced succession between the innate and adaptive immune responses [84,93].

Indeed, the key role of neutrophils in inducing the activation and cytokine release of other inflammatory cells is widely recognized [2], although the types of cytokines released by these cells have been questioned, while macrophages have been indicated as a direct source of regulatory cytokines [91].

A critical phase in wound healing is the transition from inflammation to tissue repair. The resolution of inflammation is a tightly regulated process, and neutrophils play an essential role by undergoing programmed cell death (apoptosis) and triggering the clearance of apoptotic cells by macrophages. As has been indicated, a particular subtype group of neutrophils emerges following defensive action as “veteran neutrophils”. These neutrophils, which have performed the effector action of phagocytosis, are able to leave the site by re-entering the vasculature, activating the so-called process of ‘‘reverse migration’’ [46], avoiding neutrophil persistence that could prolong the inflammatory phase and compromise the transition to the proliferative phase of healing (Figure 3A).

Exhausted neutrophils release factors such as VEGF and transforming growth factor-beta (TGF-β) that promote angiogenesis and fibroblast proliferation [26]. These findings highlight the occurrence of a reconstructive/reparative neutrophil subtype in wound healing. It is important to considerer that neutrophils are the only cells in the body capable of releasing (matrix metalloproteinase-9) MMP-9 free of its endogenous inhibitor, delivering highly active MMP-9 to angiogenic sites [17].

In the reparative stage of wound healing, a key event is represented by the conversion of inflammatory M1-type macrophages into reparative M2 macrophages. This switch is mediated in response to the downregulation of IL-10 and upregulation of IL-4 and IL-13, as well as in response to specific mediators [95] (Figure 1C). Some of these factors also appear to be produced by neutrophils, which have therefore been implicated in the shift of macrophage phenotypes from M1 to M2, identifying a further specific neutrophil subtype. However, the real neutrophil production of these cytokines has been questioned [91]. Indeed, experiments performed in mice have confirmed the participation of neutrophils in macrophage polarization, but this scenario has not yet been observed in humans.

All of these findings indicate the direct involvement of different neutrophil subtypes in coordinating the switch from inflammation to tissue repair. Otherwise, the importance of neutrophils for effective wound repair is confirmed by the observation that neutropenic individuals often have difficulty healing wounds [22].

### 4.2. Altered Wound Healing

Neutrophil heterogeneity is a relatively new concept that has emerged in recent years, and understanding the neutrophil heterogeneity in wound healing could have significant clinical implications [41].

In chronic wounds, such as diabetic ulcers, the healing process is primarily characterized by the chronicity of inflammatory conditions (Figure 3B). The neutrophil recruitment, function and drainage are impaired, leading to prolonged inflammation and delayed healing [11]. In particular, the dysregulation of neutrophil heterogeneity contributes to impaired wound healing, as certain subtypes of neutrophils may be less effective at clearing infections, fail to effectively clear bacteria or produce excessive ROS, resulting in tissue damage (Figure 3B). In this context, an increase in the physiological subpopulation of aged neutrophils could occur [96]. Moreover, it has been observed that in some chronic wounds, the neutrophil subtypes may adopt mainly an immunosuppressive phenotype, suppressing the function of other immune cells such as macrophages and T cells and impairing the overall healing process [54,97,98].

Targeting specific neutrophil subpopulations or their signaling pathways could provide therapeutic benefits. For example, strategies aimed at improving the recruitment or function of N2-like neutrophils could help accelerate wound healing. A better understanding of the signals that promote neutrophil plasticity (in particular, the switch from the inflammatory to reparative phenotypes) could lead to therapies that enhance tissue repair while limiting excessive inflammation or fibrosis. Since neutrophils and macrophages interact closely in the wound healing process, manipulating these interactions could improve healing outcomes.

### 4.3. Wound Healing in Aging

Studies dating back to 1916 have recognized that skin wound healing in the elderly is delayed, as has been further confirmed by substantial clinical and experimental evidence [99], although this concept has been challenged more recently. In fact, several studies have highlighted that wound healing in the elderly is delayed but not defective, and the poor healing of chronic wounds in the elderly can be more often attributable to comorbid conditions, rather than age alone [100].

Neutrophils have been observed to lose their efficiency with aging. Neutrophils show an impaired ability to phagocytose and destroy pathogens, which compromises the immune response [15,101]. Furthermore, the neutrophil presence at the end of the inflammatory stage of wound healing defines the condition of chronic non-healing wounds, such as in diabetic foot ulcers [3,6]. An altered amount of ROS in wound healing in the elderly has been observed, impacting not only the effectiveness in terms of killing pathogens, but also potentially inducing tissue damage. In the case of excessive ROS production, the NET production in neutrophils is impaired, altering the protective efficacy of the NETs, which conversely can induce tissue/organ disease [3,15,101].

Neutrophils increase their heterogeneity during aging [3]. This heterogeneity encompasses differences in function, phenotype and behavior, which can affect the immune response and contribute to the increased susceptibility to infections, chronic inflammation and other age-related diseases commonly observed in elderly populations [3].

The mechanisms by which neutrophil heterogeneity can modulate wound healing, especially in pathological conditions, are poorly investigated, along with the development of therapies specifically targeting neutrophil function and regulation. For example, in conditions such as diabetes, the dysregulation of neutrophil heterogeneity may contribute to impaired wound healing [41,97]. From a therapeutic perspective, several animal studies have highlighted the therapeutic potential of heterogeneity modulation to improve wound healing. For example, the use of anti-inflammatory agents [102], and the inhibition of neutrophil-derived ROS and proteases [103] have been shown to reduce oxidative stress and promote angiogenesis [104].

However, this field of investigation would benefit greatly from the characterization of biomarkers that are effective in distinguishing, for example, pro-inflammatory neutrophils from repairing ones, both in healthy subjects and especially in patients who suffer from an alteration of the wound healing process, such as diabetics or elderly subjects. The effective subsetting of neutrophils could actually allow the use of pharmacological or even personalized therapies to restore normal healing capacity.

## 5. Future Direction

Neutrophils are traditionally viewed as a relatively homogeneous population with a primary role in the innate immune response, particularly in fighting bacterial infections. A growing body of evidence indicates that the possession of a special “plasticity” also allows phenotypic and functional heterogeneity in the context of wound healing. In this scenario, their roles evolve over time, from pathogen clearance and inflammation to tissue repair and resolution of inflammation [2,34,41]. Therefore, better understanding the nature of neutrophil subtypes and their interactions with other immune cells has become an interesting challenge in providing opportunities to improve wound healing, particularly in patients with chronic or non-healing wounds.

From the summary of the large, but not so updated and homogeneous mass of information and discussions on neutrophil heterogeneity, it emerges that neutrophils could be operationally classified into three significant functional classes that we can call defensive neutrophils, helper neutrophils and regenerative neutrophils.

The defensive neutrophil class expresses all of the classical immunodefensive performances widely described to belong to these cells. Helper neutrophils became essential to assist the functional expression of T lymphocytes and especially B lymphocytes in their antibody production and in their memory cell creation [105]. Finally, regenerative neutrophils appear to be critical to initiate the wound healing reconstructive phase, as shown by the content of some granules in the cytoplasm, containing factors addressing the early reconstructive phase of the extracellular matrix and organizing the intervention of T lymphocytes in this phase as well.

These three classes are not represented by three different lineages of neutrophils, but they are all traceable to one type of immature neutrophil capable of differentiating in situ, following the different local stimuli and different secreted cytokines that occur in the place where it operates [32]. This identifies the perspective that the three different classes are masks and not rigid lineages of neutrophils, opening an opportunity to intervene in modifying the stimuli that allow neutrophils to belong to one class rather than another.

Neutrophil heterogeneity can well represent a key aspect of the immune response that allows neutrophils to adapt to a wide range of pathogenic challenges and tissue microenvironments, allowing an opportune and rapid response when tissue conditions quickly change and balance, becoming key factors in both defensive and healing processes [32].

The real clinical challenge remains in recognizing the neutrophil aspects using reliable markers and understanding how neutrophil heterogeneity can be modulated to organize appropriate therapeutic interventions in impaired wound healing.

### Implications for Personalized Medicine

Neutrophil heterogeneity may represent both challenges and opportunities in the field of personalized medicine. Understanding neutrophil heterogeneity can aid in stratifying patients based on their immune profiles. This could help identify specific subsets of neutrophils associated with different disease outcomes, allowing for more accurate prognostication.

Neutrophil characteristics, such as specific surface markers or functional assays, could serve as potential biomarkers for disease activity or treatment responses. This could enable a more tailored treatment approach for individual patients. Personalized therapies could be designed to modulate neutrophil function in diseases such as SLE, diabetes or aging-altered wound healing and enhance neutrophil responses in infectious diseases.

Continued research in this area is essential to unlock the full potential of neutrophils in wound healing disorders.

## Figures and Tables

**Figure 1 biomedicines-13-00694-f001:**
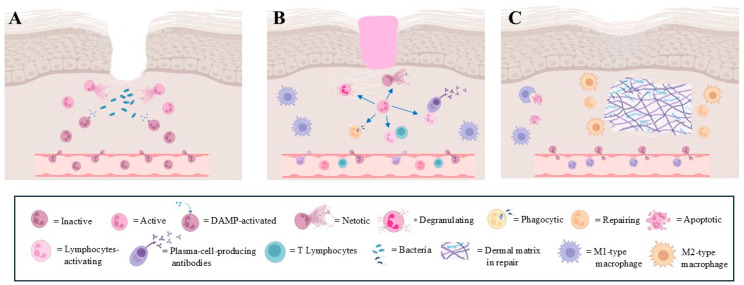
Main steps of physiological wound healing. Heterogeneous role of neutrophils is evidenced. (**A**): recruitment, tissue invasion and activation of neutrophils following wound; (**B**): different neutrophil responses during defensive phase in wound healing. Each modality of defensive response is represented by different type of neutrophil; (**C**): final step of wound healing, where specific type of neutrophil participates together with M2-type macrophages in restoring extracellular matrix integrity (images created with https://www.biorender.com (Free access—accesses repeated to build the images)).

**Figure 2 biomedicines-13-00694-f002:**
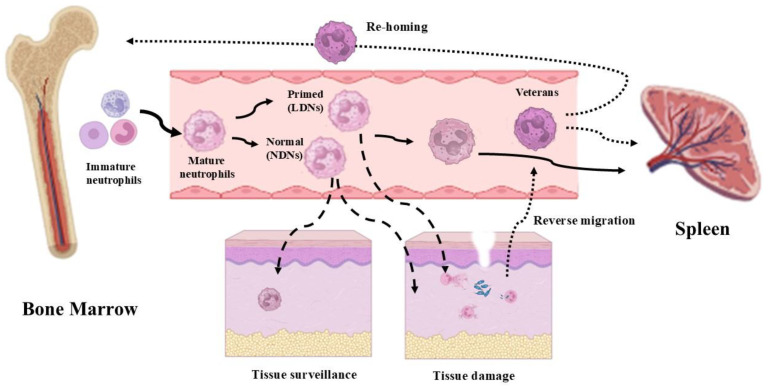
The circulation and destiny of neutrophils. Mature neutrophils, derived from the bone marrow, can remain in circulation and then be eliminated after a certain period of time in the spleen (continuous lines). Normal-density neutrophils (NDNs) can be involved both in tissue surveillance and in infection defense actions. Low-density neutrophils (LDNs) are properly involved in infection defense actions (dashed line). Recently, a new type of neutrophil (veterans), capable of reverse migration, has been observed. An interesting aspect of veteran neutrophils is that they can be either eliminated by the spleen or directly returned to the bone marrow (dotted lines) (images created with https://www.biorender.com (Free access—accesses repeated to build the images)).

**Figure 3 biomedicines-13-00694-f003:**
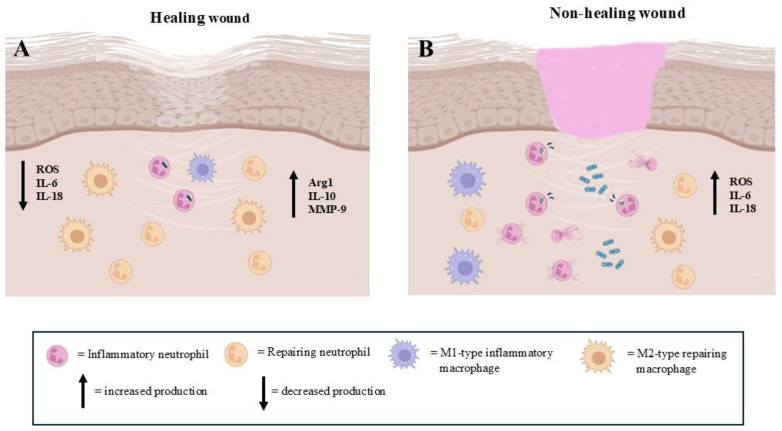
The presence of pro-inflammatory and pro-repairing neutrophils in the regenerative step of extracellular matrix repair events. (**A**): The prevalence of repairing neutrophils allows for correct wound healing. (**B**): The high number of pro-inflammatory neutrophils impairs the wound healing process (images created with https://www.biorender.com (Free access—accesses repeated to build the images)).

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
