# Peer review of "Neutrophil Heterogeneity in Wound Healing"

_biomedicines, 2025, doi:10.3390/biomedicines13030694_

Round 1

Reviewer 1 Report

Comments and Suggestions for Authors

Dear Authors,

there are my comments on your article.

  1. You did not provide an answer to the question you've put to the title. It seems that part 4.2 was intended to clarify this subject, but there are generalities only. No direct information on the role of different subtypes of neutrophils in wound healing.
  2. On line 208 you're referring to the Fig.2, however, it bears no mentions on T- or NK-cells.
  3. I don't understand, what did you mean on lines 413-414/ It sounds like you make no differences between quiescent and exhausted neutrophils, while these subpopulations are different.
Comments on the Quality of English Language
  1. The text contains a lot of grammar mistakes (I've roughly counted 30). Sometimes they change the meaning of the text (e.g. etherogeneity of neutrophils supose they are able to produce ethers; sunset neutrophils are unknown to me; etc).
  2. Some acronyms are introduced more than once (e.g. ROC, NETs and other).
  3. Some sentences are incomplete and therefore, hard to understand (lines 90-91;117-119;208-211; 299-302; 327-328;342-344; 448-500).

Author Response

In the revised text: 
Yellow = deleted sentences
Green = inserted sentences
Comment 1:  You did not provide an answer to the question you've put to the title. It seems that part 4.2 was intended to clarify this subject, but there are generalities only. No direct information on the role of different subtypes of neutrophils in wound healing.
Response 1:  The title has been changed in order to clarify the subject of the review that is, however, a topic “in progress” poorly investigated. This is the reason why few direct information about the role of different subtypes of neutrophils in wound healing are presented. We hope that this work will arouse interest in this topic.

Comment 2:  On line 208 you're referring to the Fig.2, however, it bears no mentions on T- or NK-cells.
Response 2:  The indication “Fig. 2” was erroneously placed. It has been shifted in a more appropriate position.

Comment 3:  I don't understand, what did you mean on lines 413-414/ It sounds like you make no differences between quiescent and exhausted neutrophils, while these subpopulations are different.
Response 3:  The Sentence has been rephrased, and now we hope that it could be more readable.

Comments on the Quality of English Language
Comment 1: The text contains a lot of grammar mistakes (I've roughly counted 30). Sometimes they change the meaning of the text (e.g. etherogeneity of neutrophils supose they are able to produce ethers; sunset neutrophils are unknown to me; etc).
Response 1:  Corrections have been made, according to referee’s suggestions

Comment 2:  Some acronyms are introduced more than once (e.g. ROC, NETs and other).
Response 2:  Acronyms have been checked and corrected based on referee’s suggestion

Comment 3:  Some sentences are incomplete and therefore, hard to understand (lines  90-91; 117-119; 208-211; 299-302; 327-328; 342-344; 448-500). 
Response 3:  We have reformulated the indicated sentences, in a tentative to improve their understanding

Reviewer 2 Report

Comments and Suggestions for Authors

This work is focused on the review of current knowledge of human neutrophils heterogeneity and the role of these cells in wound healing. This subject seems to be important to understand the impact of neutrophil subtypes in wound healing and might be helpful in developing new approaches to the treatment of chronic or non-healing wounds.
In general, the article can be published in the Journal of Biomedicines after revision.
Major aspect.
The issue of heterogeneity of neutrophils and possible impact of derangement of a select neutrophil population on human health was discussed earlier (Gallin JI. Human neutrophil heterogeneity exists, but is it meaningful? Blood. 1984 
May;63(5):977-83). At this moment, there is numerous studies indicating heterogeneity of neutrophils. The authors provide a summary of some evidence. Although, the manuscript was a well-structured and well-written, I would like to recommend adding  a scheme (Figure or Table), summarizing evidence for neutrophils heterogeneity (differences in receptor expression, functional activity,  expression of various factors, a type of cell death, the origin of neutrophil heterogeneity (including local stimuli, which influence the subtype of neutrophils) etc).
Minor aspect.
The text contains numerous typos (some of them marked in the text). Please, check carefully.

Author Response

In the revised text: 
Yellow = deleted sentences
Green = inserted sentences
Major aspect.
Comment 1:  The issue of heterogeneity of neutrophils and possible impact of derangement of a select neutrophil population on human health was discussed earlier (Gallin JI. Human neutrophil heterogeneity exists, but is it meaningful? Blood. 1984   May;63(5):977-83). 
Response 1:  A sentence has been added, and the reference has been cited

Comment 2:  At this moment, there is numerous studies indicating heterogeneity of neutrophils. The authors provide a summary of some evidence. Although, the manuscript was a well-structured and well-written, I would like to recommend adding a scheme (Figure or Table), summarizing evidence for neutrophils heterogeneity (differences in receptor expression, functional activity,  expression of various factors, a type of cell death, the origin of neutrophil heterogeneity (including local stimuli, which influence the subtype of neutrophils) etc).
Response 2:  Following the referee’s suggestion, a Table has been inserted in the text.

Minor aspect.
Comment 1:  The text contains numerous typos (some of them marked in the text). Please, check carefully.
Response 1:  The typos have been corrected, and the English language has been revised.

Reviewer 3 Report

Comments and Suggestions for Authors
  • The reverse migration of neutrophils to the bone marrow and their role in "training" myeloid progenitors through epigenetic reprogramming is still under investigation and lacks definitive experimental proof in humans. Additional references supporting this hypothesis would be beneficial.
  • The manuscript mentions the role of low levels of NETs in promoting keratinocyte proliferation via NF-κB. However, it does not sufficiently address the cytotoxic effects of excessive NETosis and its contribution to chronic wounds and fibrosis. A more balanced discussion reflecting this duality is needed.
  • The claim that macrophage depletion does not affect scar formation contradicts much of the literature, which describes macrophages as critical for extracellular matrix remodeling. If a specific study supports this claim, it should be explicitly cited.
  • The manuscript applies the N1/N2 neutrophil paradigm, based on murine models, to humans. However, this classification lacks direct evidence in human studies. A disclaimer highlighting the limitations of translating murine data to human neutrophil biology is necessary.
  • The proangiogenic role of neutrophils, particularly through VEGF secretion and MMP-9 release, is discussed but lacks a clear mechanistic explanation. Are specific neutrophil subsets primarily responsible for this function? Clarification would strengthen the discussion.
  • The discussion on neutrophil heterogeneity does not explore how metabolic changes (e.g., glycolysis vs. oxidative phosphorylation) influence different neutrophil subtypes. This emerging field should be acknowledged.
  • While future directions mention the potential for modulating neutrophil function in wound healing, no specific therapeutic strategies are proposed. The manuscript would benefit from discussing Neutrophil-targeting drugs (e.g., IL-8/CXCR2 inhibitors for chronic inflammation), Neutrophil reprogramming via metabolic interventions, Potential biomarkers to distinguish pro-inflammatory vs. pro-repair neutrophils
  • Some typos and grammatical errors need correction, such as: "etherogeneity" (Line 64) should be "heterogeneity." "partecipate" (Line 57) should be "participate."
  • What are the key transcription factors and signaling pathways that determine neutrophil subtype differentiation?
  • Does the aging-induced decline in neutrophil function correlate with specific transcriptional changes, or is it primarily due to cellular exhaustion?
  • How can insights into neutrophil plasticity be applied in personalized medicine for wound healing disorders?

Author Response

In the revised text: 
Yellow = deleted sentences
Green = inserted sentences

Comment 1: The reverse migration of neutrophils to the bone marrow and their role in "training" myeloid progenitors through epigenetic reprogramming is still under investigation and lacks definitive experimental proof in humans. Additional references supporting this hypothesis would be beneficial. 
Response 1: Additional reference on reverse migration observed in patients has been added “Buckley et al., 2006”.

Comment 2:  The manuscript mentions the role of low levels of NETs in promoting keratinocyte proliferation via NF-κB. However, it does not sufficiently address the cytotoxic effects of excessive NETosis and its contribution to chronic wounds and fibrosis. A more balanced discussion reflecting this duality is needed.
Response 2:  Toxic effect of NETosis has been reported according to reviewer’s suggestion

Comment 3:  The claim that macrophage depletion does not affect scar formation contradicts much of the literature, which describes macrophages as critical for extracellular matrix remodeling. If a specific study supports this claim, it should be explicitly cited.  
Response 3:  The macrophage depletion role has been investigated both in animal model (ref n. 20, “Lucas et al., 2010 - Differential roles of macrophages in diverse phases of skin repair-”) and in neutropenic patients (ref n. 21, “Nathan, 2006 - Neutrophils and immunity: challenges and opportunities”). The sentence in the text has been modified.

Comment 4:  The manuscript applies the N1/N2 neutrophil paradigm, based on murine models, to humans. However, this classification lacks direct evidence in human studies. A disclaimer highlighting the limitations of translating murine data to human neutrophil biology is necessary. 
Response 4:  In our paper we specified that N1 and N2 subset was observed only in the mouse model, and we did not hypothesize any similarity with human subset.

Comment 5:  The proangiogenic role of neutrophils, particularly through VEGF secretion and MMP-9 release, is discussed but lacks a clear mechanistic explanation. Are specific neutrophil subsets primarily responsible for this function? Clarification would strengthen the discussion. 
Response 5:  The authors have added a sentence about angiogenic mechanism and pro-angiogenic population

Comment 6:  The discussion on neutrophil heterogeneity does not explore how metabolic changes (e.g., glycolysis vs. oxidative phosphorylation) influence different neutrophil subtypes. This emerging field should be acknowledged…. 
Response 6:  Authors thank the reviewer for the suggestion about metabolic changes in neutrophil. We added a small paragraph, that we hope could complete our reviewer (line 356-378 – correction version).

Comment 7:  While future directions mention the potential for modulating neutrophil function in wound healing, no specific therapeutic strategies are proposed. The manuscript would benefit from discussing Neutrophil-targeting drugs (e.g., IL-8/CXCR2 inhibitors for chronic inflammation), Neutrophil reprogramming via metabolic interventions, Potential biomarkers to distinguish pro-inflammatory vs. pro-repair neutrophils.
Response 7:  We have added a small paragraph to address the reviewer observation (line 561-566 – correction version)

Comment 8:  Some typos and grammatical errors need correction, such as: "etherogeneity" (Line 64) should be "heterogeneity." "partecipate" (Line 57) should be "participate."
Response 8:  Corrections have been made 

Comment 9:  What are the key transcription factors and signaling pathways that determine neutrophil subtype differentiation? 
Response 9:  Authors thank the reviewer for the suggestion about transcription factors and signaling pathways in neutrophil. We added a paragraph, that we hope could complete our reviewer (line 380-425 – correction version)

Comment 10:  Does the aging-induced decline in neutrophil function correlate with specific transcriptional changes, or is it primarily due to cellular exhaustion? 
Response 10:  A paragraph about the specific transcriptional changes in neutrophil aging process has been added (line 413-425 – correction version)

Comment 11:  How can insights into neutrophil plasticity be applied in personalized medicine for wound healing disorders?  
Response 11:  A paragraph about the relationship between personalized medicine and neutrophil heterogeneity has been added (line 612-624 – correction version)

Round 2

Reviewer 1 Report

Comments and Suggestions for Authors

Dear authors,

I see you've worked on the text, nevertheless I still have some comments. I'm listing them by the way they appear in the text, not by their importance.

  1. You introduce an acronym PMNs on line 80, where the neutrophils only are mentioned. It is not clear, why 'PMNs' refers to 'neatrophils'.
  2. The sentence on lines 110-114 seems to be inconsistent.
  3. M2 macrophage occur on line 137 without any explanation. Please, add a few word on the macrophage subtypes and shortly describe roles of M1 and M2 cells.
  4. You first mention 'veteran neutrophils' on line 200 without any explanation what does it mean. Please, add an explanation.
  5. You're mentioning LD neutrophils on lines 217, 218, 228. Nothing is said about neutrophils with another density. Then, on line 274 and further, you're introducing LDN and HDN and use these abbreviations. Please, use single acronym for single cell population.
  6. You first mention 'standard neutrophils' on line 282 and again do not offer any explanation of the term. The description of  'standard neutrophils' appears on lines 302-304. Please, correct.
  7. Acronym SLE is introduced twice on one page (lines 285 and 296).
  8. Subsection 'Neutrophils heterogeneity in wound healing' contains no information on any neutrophils' subtypes, described earlier. It once more describes the role of neutrophils in general in wound healing. As this appears to be one of the most important part of the whole text, it should be rewritten completely. The exact role of each subtype in wound healing should be described.
  9. In subsection 5 you're saying that all neutrophil subtypes have a dynamic nature (lines 581, 582) which slightly contradicts the descriptions of those cells given before.

Summarizing my comments I'd suggest thoroughly rewrite the text concentrating on your main idea.

Comments on the Quality of English Language

The text still has enormous number of typos!!! Please, use Word grammar check at least!!! It seems disrespectful to send such careless text to the publishing! And, please, get rid of all 'etherogeneity' in the text!!!!!

Author Response

In the revised text: 
Yellow = deleted sentences
Green = inserted sentences

Comment 1: You introduce an acronym PMNs on line 80, where the neutrophils only are mentioned. It is not clear, why 'PMNs' refers to 'neatrophils'.
Response 1: We agree with referee note; the acronym is confusing and unnecessary. It has been deleted.

Comment 2: The sentence on lines 110-114 seems to be inconsistent.
Response 2: an explanatory sentence has been added (line 114-116 – corrected version)

Comment 3:  M2 macrophage occur on line 137 without any explanation. Please, add a few word on the macrophage subtypes and shortly describe roles of M1 and M2 cells.
Response 3: A paragraph about the role of M1 and M2 macrophages has been added (line 94-99 – corrected version)

Comment 4: You first mention 'veteran neutrophils' on line 200 without any explanation what does it mean. Please, add an explanation.
Response 4:  an explanatory sentence has been added (line 204-206 – correction version)

Comment 5:  You're mentioning LD neutrophils on lines 217, 218, 228. Nothing is said about neutrophils with another density. Then, on line 274 and further, you're introducing LDN and HDN and use these abbreviations. Please, use single acronym for single cell population.
Response 5: the use of the same acronym for single population has been adopted.

Comment 6:  You first mention 'standard neutrophils' on line 282 and again do not offer any explanation of the term. The description of  'standard neutrophils' appears on lines 302-304. Please, correct.
Response 6: The first mention of “standard neutrophils” has been removed, as it was unexplained and confusing. 

Comment 7:  Acronym SLE is introduced twice on one page (lines 285 and 296).
Response 7:  The correction has been made

Comment 8: Subsection 'Neutrophils heterogeneity in wound healing' contains no information on any neutrophils' subtypes, described earlier. It once more describes the role of neutrophils in general in wound healing. As this appears to be one of the most important part of the whole text, it should be rewritten completely. The exact role of each subtype in wound healing should be described.
Response 8:  The role of different subtypes of neutrophils in wound healing has been better identified  

Comment 9:  In subsection 5 you're saying that all neutrophil subtypes have a dynamic nature (lines 581, 582) which slightly contradicts the descriptions of those cells given before.
Response 9: the term dynamic has been removed as unnecessary and confusing. 

Summarizing my comments I'd suggest thoroughly rewrite the text concentrating on your main idea.
We thanks the reviewer for the suggestions that allow a partial rearrange of the text

Comments on the Quality of English Language
The text still has enormous number of typos!!! Please, use Word grammar check at least!!! It seems disrespectful to send such careless text to the publishing! And, please, get rid of all 'etherogeneity' in the text!!!!!

Response: We apologize for the happened due to Word text editing program not working. We have now checked the typos with an alternative text editor. The corrections have been highlighted

Reviewer 2 Report

Comments and Suggestions for Authors

The manuscript can be accepted.

Author Response

Comment: The manuscript can be accepted.
Response: The authors are grateful to reviewer for the valuable revision of our work

Reviewer 3 Report

Comments and Suggestions for Authors

Thanks for incorporating the comments suggested in the revised manuscript.

Author Response

Comment: Thanks for incorporating the comments suggested in the revised manuscript.
Response: The authors are grateful to referee for the valuable review of our work

Round 3

Reviewer 1 Report

Comments and Suggestions for Authors

Dear authors,

thank you for the work you've made on the manuscript. It seems much better now. I'm fully satisfied.